# Velocity Analysis Using Separated Diffractions for Lunar Penetrating Radar Obtained by Yutu-2 Rover

**Chao Li** [1,2,3] **and Jinhai Zhang** [1,2,*]

1    Key Laboratory of Earth and Planetary Physics, Institute of Geology and Geophysics,
     Chinese Academy of Sciences, Beijing 100029, China; superlee@mail.iggcas.ac.cn
2    Innovation Academy of Earth Science, Chinese Academy of Sciences, Beijing 100029, China
3    College of Earth and Planetary Sciences, University of Chinese Academy of Sciences, Beijing 100029, China
*    Correspondence: zjh@mail.iggcas.ac.cn

**Abstract:** The high-frequency channel of lunar penetrating radar (LPR) onboard Yutu-2 rover successfully collected high quality data on the far side of the Moon, which provide a chance for us to detect the shallow subsurface structures and thickness of lunar regolith. However, traditional methods cannot obtain reliable dielectric permittivity model, especially in the presence of high mix between diffractions and reflections, which is essential for understanding and interpreting the composition of lunar subsurface materials. In this paper, we introduce an effective method to construct a reliable velocity model by separating diffractions from reflections and perform focusing analysis using separated diffractions. We first used the plane-wave destruction method to extract weak-energy diffractions interfered by strong reflections, and the LPR data are separated into two parts: diffractions and reflections. Then, we construct a macro-velocity model of lunar subsurface by focusing analysis on separated diffractions. Both the synthetic ground penetrating radar (GPR) and LPR data shows that the migration results of separated reflections have much clearer subsurface structures, compared with the migration results of un-separated data. Our results produce accurate velocity estimation, which is vital for high-precision migration; additionally, the accurate velocity estimation directly provides solid constraints on the dielectric permittivity at different depth.

**Keywords:** lunar penetrating radar; plane-wave destruction; focusing analysis; diffractions; velocity analysis

## 1. Introduction

The lunar subsurface material and geological structure recorded the early evolution history of the Moon. In the past half century, several radar detections have been carried out to study subsurface structures of the Moon, such as Earth based radar [1,2], orbital radar [3–5], and lunar penetrating radar (LPR) [6–8]. The orbital radar can penetrate deep into the ground because of using low-frequency radar waves and can obtain a global coverage; however, it has relatively low spatial resolution on the local details of stratifications [9]. In contrast, the LPR emits and receives the reflected high-frequency radar waves directly from the lunar subsurface anomalies, such as reflection layers and small diffractors, thus can detect local details of subsurface structures (Figure 1) [6]. Chang'E-3 landed in the Imbrium basin on the near side of the Moon in 2013, and the Yutu rover performed the first roving detection by LPR on an extraterrestrial planet [7]. There are two channels of the LPR payload: Channel 1 works with a central frequency of 60 MHz, which is to detect the deep subsurface structures with a resolution of 10 m within the depth of 500 m, and Channel 2 works with a central frequency of 500 MHz, which is to detect the shallow subsurface structures and lunar regolith with a resolution of 0.3 m within the depth of 50 m [6]. On 3 January 2019, Chang'E-4 landed in the Von Kármán crater on the farside of the Moon [8]. The Yutu-2 rover was equipped with similar LPR payloads as the Yutu rover. The LPR onboard Yutu-2 rover had accumulated a lot of data within its current

traveling distance of ~600 m, after working 26 lunar days. The LPR data revealed that the geological sediments consist of several ejecta layers that are from adjacent craters and multi-episode volcanic eruptions [8], which clearly exhibits the transportation history of lunar shallow materials.

The model of dielectric permittivity is important for high accuracy imaging and for estimating the material composition of lunar subsurface. The velocity of radar propagation within the lunar rocks and regolith can be converted from the dielectric permittivity as follows,

$$v = \frac{c}{\sqrt{\varepsilon_r}},\qquad(1)$$

where $v$ is the velocity of electromagnetic waves, $\varepsilon_r$ is the relative dielectric permittivity and $c$ is the speed of light in vacuum [10]. With this approximation, we can directly extend many sophisticated methods developed in the seismic exploration fields, such as depth migration [7] and velocity analysis, to image and identify complex structures detected by the LPR. Currently, there are two approaches for building up lunar velocity models: (1) one-dimensional empirical relations derived from statistical method based on measurements on Apollo lunar samples [7,8,11,12]; (2) hyperbola fitting on identified diffractions [13–16]. The 1D layered model assumption was only valid for the situation that the subsurface velocity variations are not changed dramatically, such as horizontal layered structure, but would have an evident error in the presence of lateral velocity variations or dipping structures. In contrast, the hyperbola fitting is effective for determining the single-point velocity at the apex of diffractions; however, this method has two inherent drawbacks: (1) the effectiveness of this method highly depends on the human subjectivity and it is time consuming for processing a large number of diffractions; (2) it is highly dependent on the signal to noise ratio of diffractions and are usually influenced by strong-energy of reflections; whereas, it is usually difficult to separate the diffractions from noisy background and strong reflections, since the diffractions have much weak energy compared with the reflections. Therefore, the hyperbola fitting method can only obtain good results when the diffractions are clear and identifiable; otherwise, it will fail to handle weak diffractions that are fully mixed with strong-energy reflections. Unfortunately, typical lunar models (ejecta and brecciated basalt) are shown to be rich in diffractions from small rocks or caves [9], which are buried under strong-energy reflections from boulders or layer boundaries that are nearly horizontal.

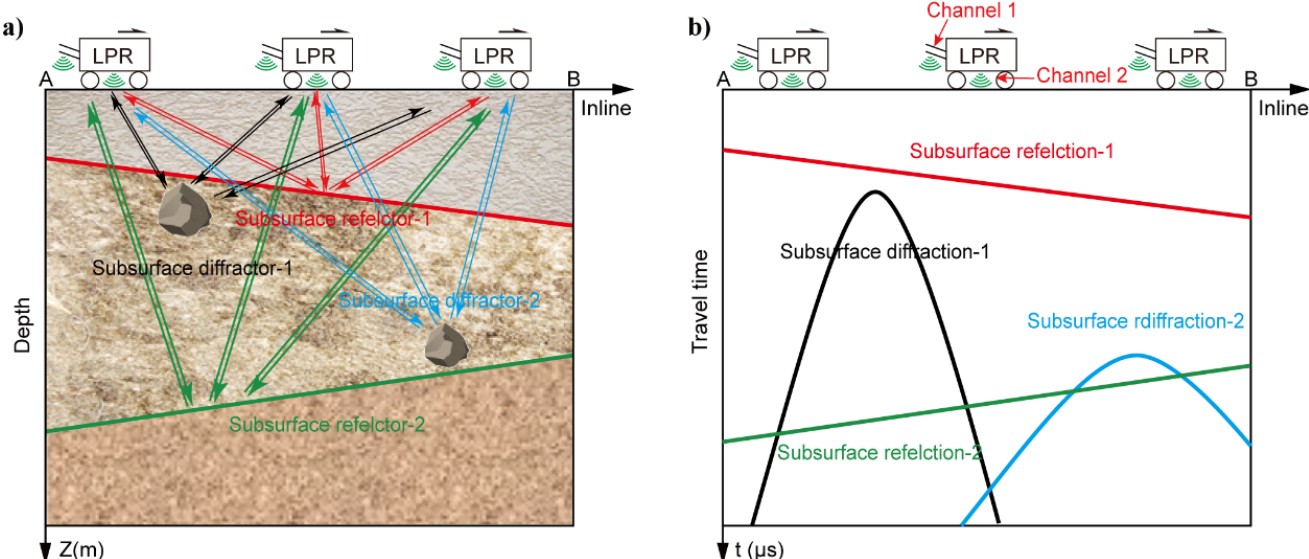

**Figure 1.** A schematic diagram of lunar penetrating radar (LPR) observation (**a**) and the recorded profile (**b**).

In the field of seismic exploration, many methods have been proposed to separate diffractions and reflections. Harlan et al. [17] proposed to separate diffractions from reflections and conducted velocity analysis by diffraction focusing. Based on local slopes, Fomel et al. [18] proposed a two-step workflow for diffraction separation and velocity analysis by plane-wave destruction (PWD) and focusing analysis. Berkovitch et al. [19] used a new time-correction formula to estimate and separate diffractions, which focused the energy of diffractions and scattered the energy of reflections. Klokov and Fomel [20] derived analytical equation for diffractions in dip-angle domain and separated diffractions and reflections by their difference in shape using Radon transform. Asgedom et al. [21] enhanced the diffractions by common-reflection-surface method and replacement-media method. Zhang and Zhang [22] used shot and opening-angle gathers to image weak diffractions. Merzlikin and Fomel [23] imaged diffractions by path-summation method and significantly reduced the computational cost by two fast Fourier transforms. Merzlikin et al. [24] divided the full wavefield into reflections, diffractions and noise by a framework consisting of Kirchhoff modeling, plane-wave destruction and path-summation integral filter. Tschannen et al. [25] used machine learning method to separate diffractions and reflections. Wang et al. [26] separated diffractions by localized rank-reduction method.

Due to the large differences between seismic and ground penetrating radar (GPR) data in the frequency components and velocity ranges, few studies have been applied to separate diffractions to GPR data. Yuan et al. [27] applied the method of Fomel et al. [18] to GPR data and the estimated velocity was shown to be consistent with the crosshole GPR velocity estimation. The framework in Fomel et al. [18] converts the diffraction separation problem into a pure signal processing task, where the GPR data are regarded as dimensionless [27], which greatly facilitates the extension of sophisticated methods that are developed for seismic data to GPR data. Economou et al. [28] used local slopes to build up a new summation weight and applied the weighted summation of constant-velocity migrated GPR profiles to achieve diffraction focusing without a velocity model.

In this paper, we first review the principle and process of Fomel et al. [18]. Secondly, we use synthetic GPR data to verify the effectiveness of this method. Thirdly, we use the PWD method to decompose Chang'E-4 LPR data into diffractions and reflections. Then, we use the local focusing analysis on diffractions to obtain a two-dimensional velocity model. Next, we convert this model into relative dielectric permittivity. Finally, we compare the imaging results after migration and draw a conclusion.

## 2. Methods

There are two key steps for the method of Fomel et al. [18]: (1) separate diffractions and reflections using the PWD method [18,29,30], which maps each trace onto neighboring traces and find out the vertical shift to minimize residual energy among the mapped trace and its adjacent traces; (2) estimate velocity using separated diffractions by focusing analysis method.

### 2.1. Plane-Wave Destruction Method

The PWD filters can be defined as the solution of local plane differential equation [29,30],

$$\frac{\partial P}{\partial x} + \sigma \frac{\partial P}{\partial t} = 0, \tag{2}$$

where $P(t, x)$ represents the wavefield, $x$ and $t$ are distance and time, $\sigma(t, x)$ represents the dominant local slopes, which describes the local variations of signals within the local window. The PWD filters formed a prediction of each trace from its adjacent traces to estimate local dominant slopes. Fomel [30] constructed a least-squares equation to estimate the local slopes,

$$C(\sigma)d \approx 0, \tag{3}$$

where $d$ represents the data, and $C(\sigma)$ denotes the operator of convolving the data with the 2D filter, which represents a transformed solution of Equation (2) [30]. We can get the local slopes by solving Equation (3) through linear iterative optimization method [30]. After local slopes are estimated, the PWD filter can remove all events with the same planer local slopes, such as reflections. Diffractions and noises are then leaved in the data, because they have non-planer local slopes. In other words, planer events mean that values of local slopes do not change within the local window and non-planer events mean that values of local slopes vary within the local window. The PWD method assumes that the reflections have similar local slopes within the local window while the diffractions have rapidly changing or even contrary local slopes within the local window [31].

### 2.2. Focusing Analysis Method

After the diffractions are separated, the focusing analysis method can be applied for velocity estimation. The varimax norm is a good measure of focusing level for separated diffractions [18,32,33]. The varimax norm is defined as

$$\phi = N \sum_{i=1}^{N} s_i^4 / \left( \sum_{i=1}^{N} s_i^2 \right)^2, \tag{4}$$

where $s_i$ are signal amplitudes inside a window of size $N$. Fomel [34] uses two continuously variable quantities to replace the varimax norm, which are solutions of the following regularized optimization problems,

$$\min_{p_i} \left( \sum_{i=1}^{N} \left( s_i^2 - p_i \right)^2 + R[p_i] \right), \tag{5}$$

$$\min_{q_i} \left( \sum_{i=1}^{N} \left( 1 - q_i s_i^2 \right)^2 + R[q_i] \right), \tag{6}$$

where $R$ represents a regularization operator, $p_i$ and $q_i$ are continuously variable quantities. By solving Equations (5) and (6), the most focused image can be found.

In summary, we first estimate the dominant local slopes of input data and use the PWD filter to separate reflections and diffractions. Second, we use velocity continuation [35,36] to get a series of migrated images of the separated diffractions. Then, we use focusing analysis method to find out the most focused image and corresponding migration velocity (i.e., root-mean-square velocity). Furthermore, we transform the root-mean-square velocity into Dix velocity and further convert it from time domain to depth domain. Next, we perform migration on the full-wavefield LPR data, the separated reflections, and the separated diffractions, respectively. Finally, we obtain clearer and focused imaging results of the separated reflections that had been migrated, compared with the migrated full-wavefield LPR data using the same velocity model.

## 3. Results

### 3.1. Synthetic Data Results

The gprMax is open source software, which uses finite-difference time-domain (FDTD) method to simulate electromagnetic wave propagation [37,38]. To verify the effectives of the PWD method and focusing analysis method, we use the gprMax to simulate a 2D synthetic GPR profile. The model consists of three dipping layers and many point-like diffractors with different relative permittivity in each layer, as shown in Figure 2. The model is 40 m in length, 20 m in depth, and the grid spacing is 0.05 m. From the top to the bottom, the background relative permittivity of each layer is 1, 3, 4, and 5, respectively. To simulate the phenomenon of weak diffractions, we set the relative permittivity of point-like diffractors

and reflectors within each layer 0.5 higher than that of the background. Additionally, for consistency with the high-frequency LPR, the central frequency of simulated antenna is 500 MHz, the sampling interval is 0.5 m, and the temporal step is 1.179 ns. Figure 2b shows the simulated GPR profile, which consists of two strong-energy reflections and lots of diffractions. For diffractions, only those in the first subsurface layer can be well identified, while those in the deeper parts are difficult to identify. If we use the original GPR profile to conduct velocity analysis by hyperbola fitting method, we can only get accurate velocity estimation for shallow parts and cannot get reliable velocity estimation for deep parts, especially in the presence of strong random noise. Therefore, we need to separate diffractions from strong-energy reflections to enhance the recognizability of weak-energy diffractions, especially for diffractions from deep depth.

Figure 3 shows the results after applying the PWD method. In Figure 3a, the two wings of the diffractions have different local slopes, which are the foundation to identify and separate reflections and diffractions. The separated reflections (Figure 3b) are clearer and more continuous compared with the original GPR profile, and only small part of energy remains around the positions of diffractors. Additionally, the diffractions are effectively separated from the strong-energy reflections, and some weak-energy diffractions that are hard to identify in the original GPR profile are also successfully identified in the separated results (see the red dashed box in Figure 3c). By comparing the original model with the separated diffractions, we find that the apexes of separated diffractions and small anomalies in each layer correspond to each other, which proves that the PWD method is powerful on separating the reflections and diffractions.

We use the separated diffractions to implement focusing analysis method. Figure 4a shows the results of the generated velocity model, which are transformed from the time domain to depth domain. The two dipping black dash lines indicate the two layer boundaries in original model. From the generated velocity model, we can observe that the focusing analysis method can obtain accurate velocity estimation from separated diffractions. Meanwhile, the migrated diffractions (Figure 4d) show that all diffractions are well focused, and the migrated reflections (Figure 4c) are much clearer than the results of migrated input GPR data (Figure 4b). Consequently, we verify by synthetic GPR data that the focusing analysis method can get accurate velocity estimation using separated diffractions.

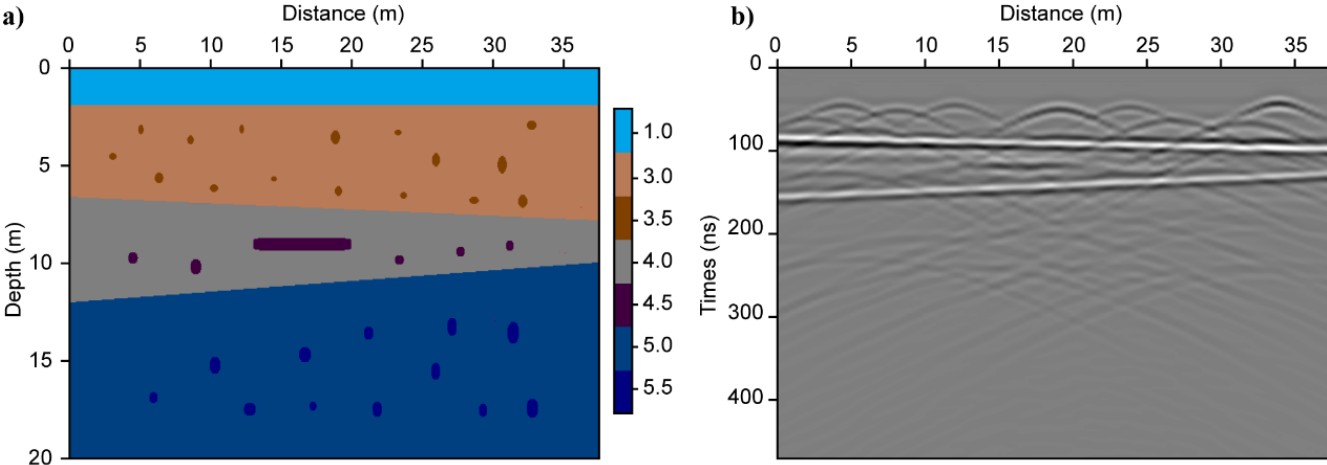

**Figure 2.** The relative dielectric permittivity model (**a**) and synthetic ground penetrating radar (GPR) data (**b**).

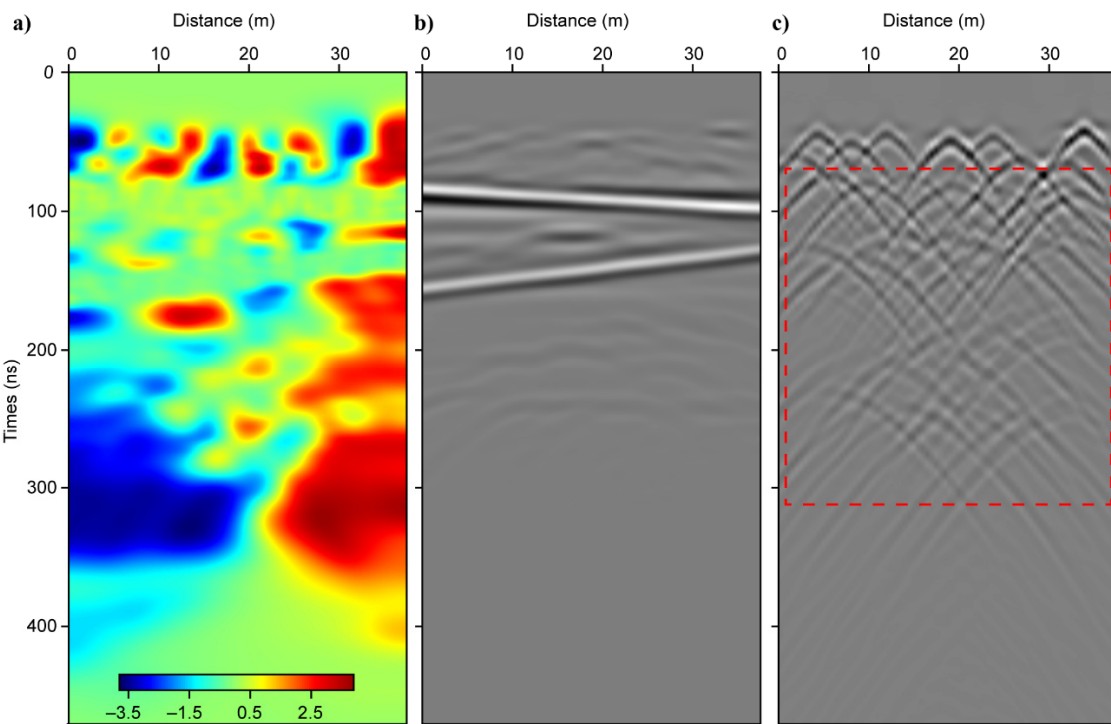

**Figure 3.** The separation results of synthetic GPR data after applying the plane-wave destruction (PWD) method. (**a**) The local slopes, (**b**) the separated reflections, (**c**) the separated diffractions. The red dashed box indicates weak-energy diffractions that are originally mixed with strong-energy reflections in the original GPR profile shown in Figure 2b.

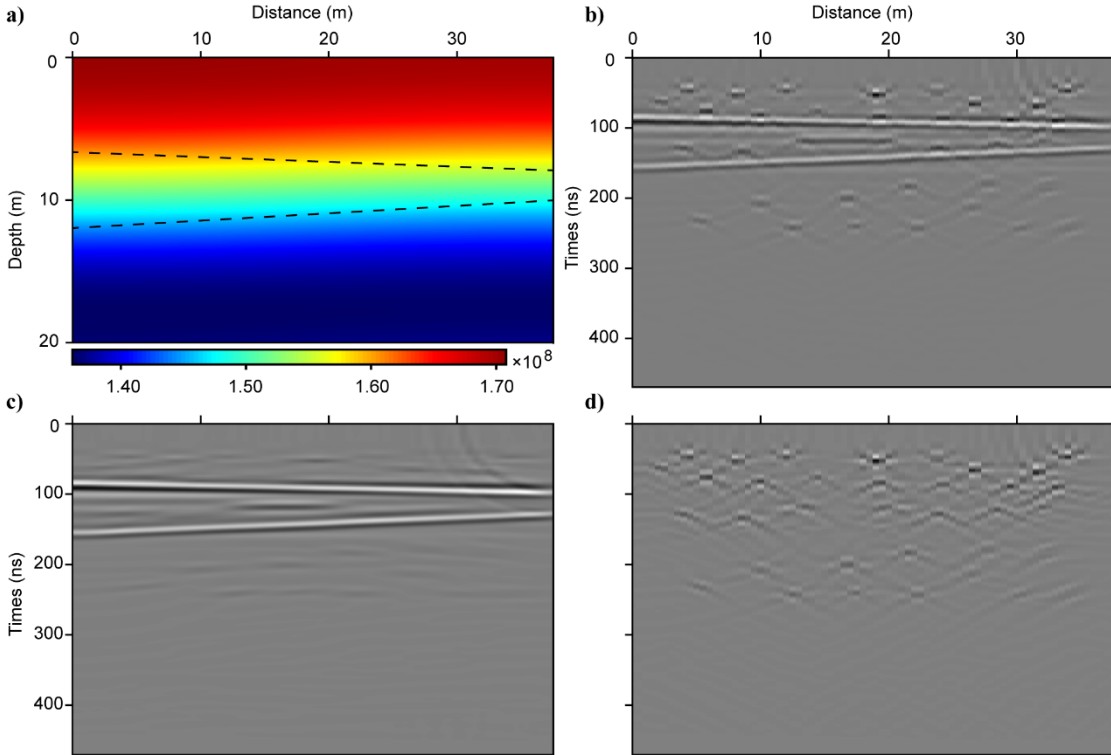

**Figure 4.** The results of focusing analysis method and time migration. (**a**) The velocity structure in the depth domain (the black dash line indicates the layer boundary), (**b**) the migrated input GPR data, (**c**) the migration results of separated reflections, (**d**) the migration results of separated diffractions.

### 3.2. LPR Data Results

The high-frequency LPR onboard Yutu-2 rover was used to detect the near-surface structure. It consists of two receiving antennas (Channel 2A and Channel 2B). The channel 2A is closer to the transmitting antenna thus has a lower signal-to-noise ratio than channel 2B. Therefore, we use the channel 2B LPR data in our experiments. The data are recorded in the first two lunar days, until 9 February 2019, with a total length of 88 m (Figure 5). The time step of channel 2B is 0.3125 ns and spatial interval is 3.65 cm. Before we apply the PWD method, the channel 2B data were processed by decoding, removing duplicative traces, removing time delay, band-pass filter, and removing background (Figure 6a,b) [7,8].

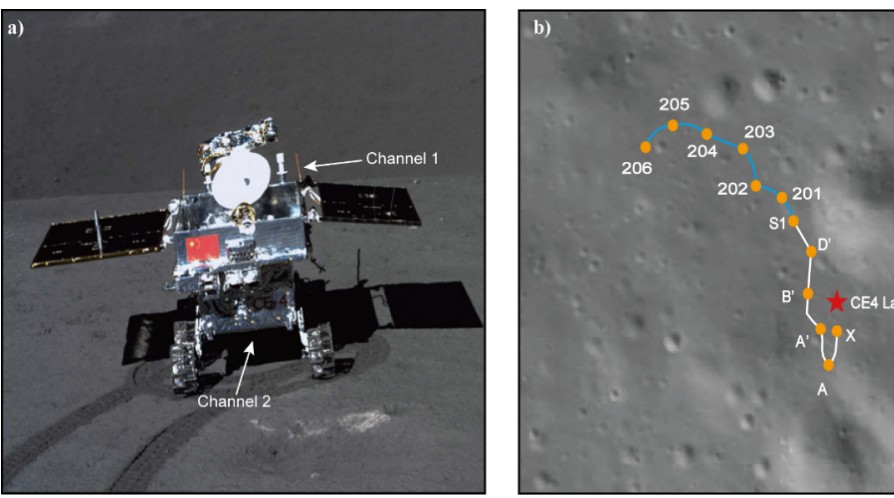

**Figure 5.** The Yutu-2 rover (**a**) and its path (**b**) in the first two lunar days.

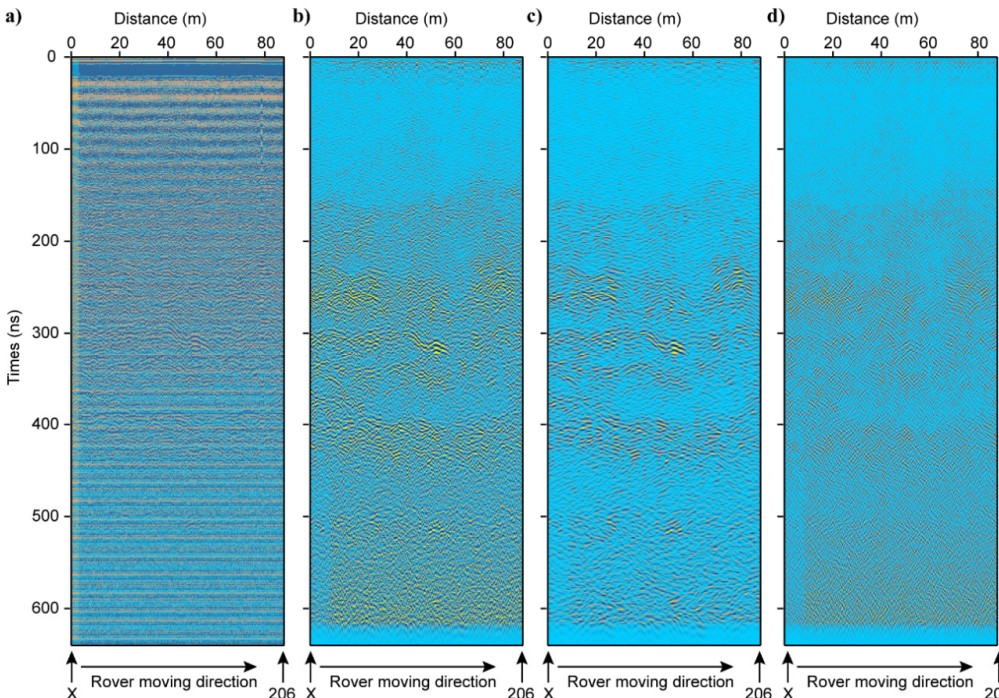

**Figure 6.** The separation results of LPR data after applying the PWD method. (**a**) The original LPR data (after decoding and removing duplicative traces), (**b**) the preliminary processed LPR data (after removing time delay, band-pass filter, and removing background), (**c**) the separated reflections, (**d**) the separated diffractions, where symbols "x" and "206" represent the starting and ending point of Yutu-2 rover shown in Figure 5, respectively.

After applying the PWD method, the preliminary processed LPR data (Figure 6b) are divided into two parts: reflections (Figure 6c) and diffractions (Figure 6d). Obviously, the separated reflections show more continuous hierarchical structures, because most hyperbola-shaped diffractions are eliminated and only small part of energy remained at the apexes of diffractions. We can clearly observe the changes of subsurface layers compared with the preliminary processed LPR data. Meanwhile, after separating from the strong-energy reflections, some weak-energy diffractions are further enhanced, which provide more available data for the velocity analysis.

Afterwards, we use separated diffractions for focusing analysis and derive a 2D velocity model, which is further converted into the relative permittivity model (Figure 7). Finally, we perform migration on the preliminary processed LPR data, the separated reflections, and the separated diffractions, respectively (Figure 8). From the estimated velocity model, we find that the lateral velocity variations are gentle and velocity interfaces are generally horizontal, which means that there are no significant kinetic effects in this region. Furthermore, the relative permittivity shows that the lunar subsurface can be roughly divided into three layers from 0 m to 50 m, according to the variations of relative permittivity. This is consistent with the results in Zhang et al. [8]. The first layer is interpreted as a lunar regolith layer from 0 m to ~15 m, whose relative permittivity is about 3~4. The second layer is a material transition zone from ~15 m to ~25 m, which gradual changes from lunar regolith to ejecta that has a relative permittivity about 4~5. The third layer is a basalt layer from ~25 m to 50 m, whose relative permittivity is about 5~6.6.

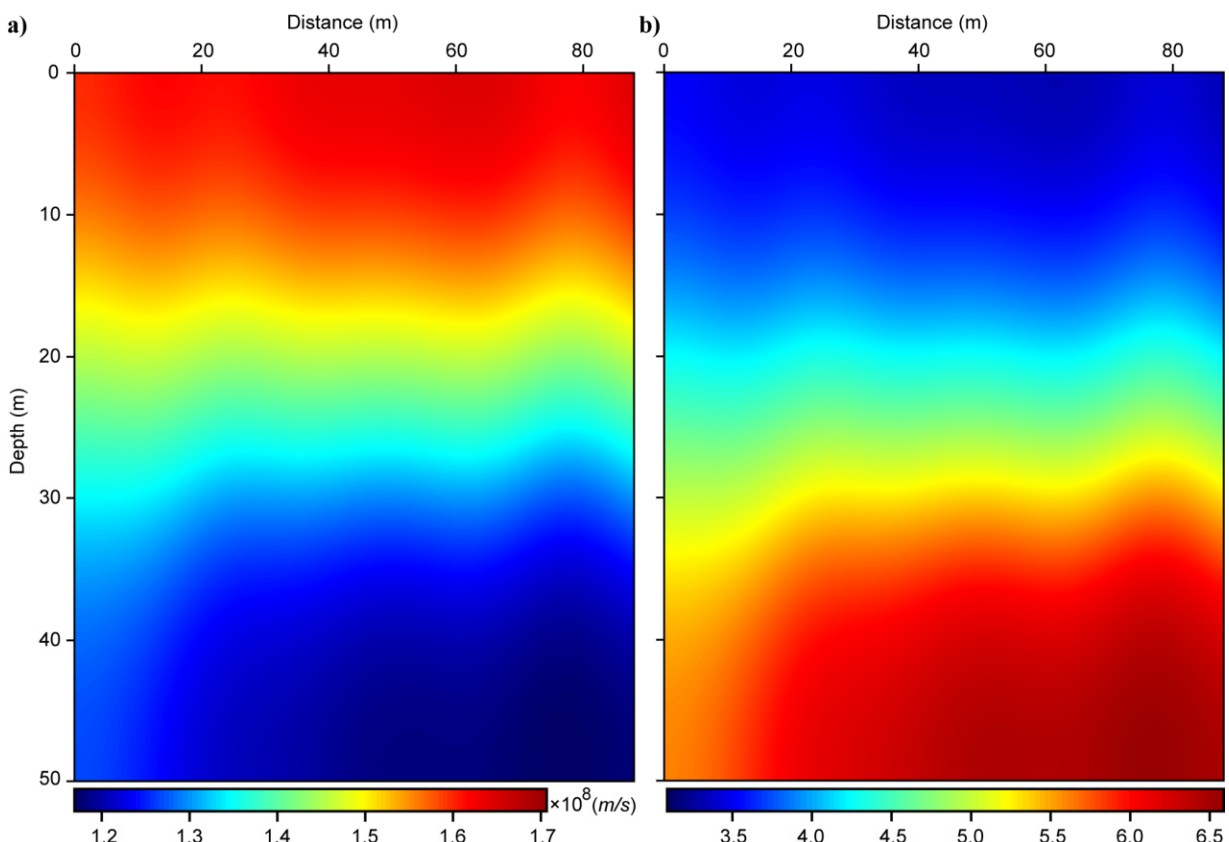

**Figure 7.** The estimated velocity model (**a**) and relative permittivity of lunar subsurface (**b**) for the LPR profile in the first two lunar days.

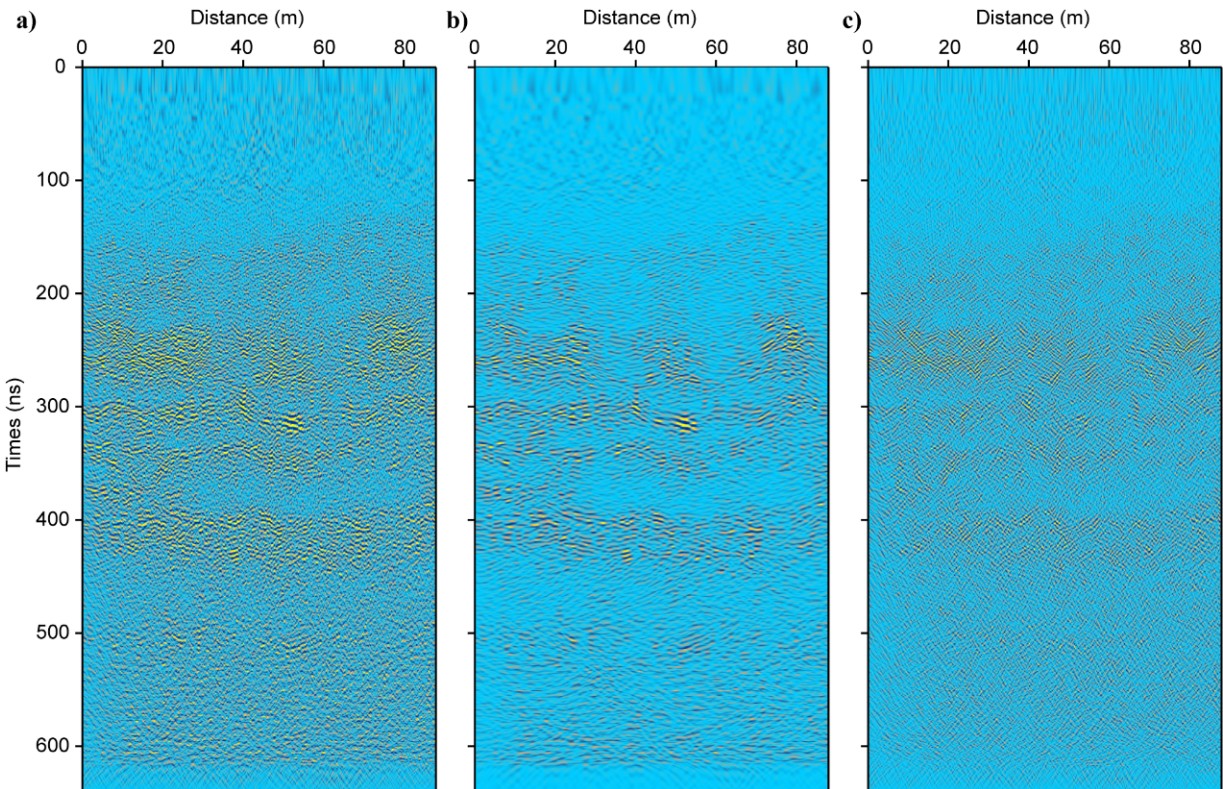

**Figure 8.** Comparison between the migrated results. (**a**) The migrated preliminary processed LPR data, (**b**) the migrated reflections after separation, (**c**) the migrated diffractions after separation.

## 4. Discussion

Diffractions from small-scale subsurface heterogeneities (such as faults and small rocks) of lunar subsurface carried rich subsurface velocity information. However, the diffractions usually have relatively weak-energy and are interfered by strong-energy reflections. Traditional hyperbola fitting method can only deal with strong-energy diffractions and does not work when the diffractions are not clear and recognizable. To perform velocity estimation from top to bottom for the LPR data, we use the PWD method to separate weak-energy diffractions from reflections, which can evidently enhance the identifiability of diffractions and avoid the influence from reflections.

In the separated reflections, we can still see some residual diffractions remained, which can influence the imaging results to some extent, but the remained energy of diffractions is relatively extraordinary low compared with unseparated data. Additionally, the residual diffractions mainly located around diffractors, where only a small part of reflections can be interfered. Certainly, in future works, we should find a more powerful method to further separate the residual diffractions, which would gain a better imaging result of reflections.

## 5. Conclusions

We introduce an effective velocity analysis method by separating reflections and diffractions of the LPR data. We illustrate the effectiveness of this method first by the synthetic GPR data. The estimated velocity model from the separated synthetic diffractions showed a good agreement with the true velocity model. We separate the reflections and diffractions of the Chang'E-4 LPR data, and the separated diffractions are used to perform velocity analysis. The method of separating diffractions has two advantages: (1) the weak-energy diffractions can be enhanced and we have more available data of diffractions for accurate velocity analysis; (2) the separated reflections can eliminate the interference of cluttered diffractions and the lunar subsurface structures are clearer after imaging. These

are important for reliable geological interpretations, especially when the reflections are highly mixed with the diffractions.

The velocity model is essential for obtaining well-focused migration results. Meanwhile, the relative permittivity converted from the velocity can provide a direct constraint on identifying the material composition [11]. We construct an accurate 2D velocity model for the lunar subsurface using the separated diffractions. Based on this model, we perform migration on the LPR data using the velocity model. The results show that lunar subsurface can be divided into three zones, which are regolith zone, the transition zone and the basalt zone, respectively, according to values of relative permittivity. We can see a more detailed structure of the lunar subsurface from the separated reflections, and such a continuous distribution of lunar subsurface layer has never been obtained by previous studies in the same research area. This method could be a powerful tool on imaging the detailed structures detected by ground penetrating radar on both lunar surface and other extraterrestrial planets.

**Author Contributions:** Conceptualization, C.L. and J.Z.; methodology, C.L. and J.Z.; software, C.L.; validation, C.L. and J.Z.; formal analysis, C.L. and J.Z.; investigation, C.L. and J.Z.; resources, C.L. and J.Z.; data curation, C.L. and J.Z.; writing—original draft preparation, C.L.; writing—review and editing, J.Z.; visualization, C.L. and J.Z.; supervision, C.L. and J.Z.; project administration, J.Z.; funding acquisition, J.Z. All authors have read and agreed to the published version of the manuscript.

**Funding:** This research was funded by the Strategic Priority Research Program of Chinese Academy of Sciences (XDA17010404) and the National Natural Science Foundation of China (41941002).

**Acknowledgments:** We thank the Supercomputing Laboratory of Institute of Geology and Geophysics, Chinese Academy of Sciences (IGGCAS) for providing computing resources and Madagascar software package for providing open-source codes (http://www.ahay.org, accessed on 3 April 2021).

**Conflicts of Interest:** The funders had no role in the design of the study; in the collection, analyses, or interpretation of data; in the writing of the manuscript, or in the decision to publish the results.

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
