# Peer review of "Velocity Analysis Using Separated Diffractions for Lunar Penetrating Radar Obtained by Yutu-2 Rover"

_remotesensing, doi:10.3390/rs13071387_

Round 1

Reviewer 1 Report

General Comments

  • Discussion and Conclusion section can be merged and shortened. There seems to be overlap in the content of these sections

Abstract

  • Explain why we need to obtain reliable dielectric permittivity model
    • I suggest moving the last line of the abstract to after the sentence on line 13.
  • What is meant by migration results? What exactly is migrating? Is it the lunar subsurface materials?
    • Why are the migration results important?

Section 1

  • Clearly emphasis the importance of doing velocity analysis. It is not clear to me why this is important, or it's relation to migration results mentioned in abstract
  • It will be useful to include a diagram or cartoon depicting the LPR and the lunar environment
    • A cartoon showing the scenario that is being considered in this paper or how the LPR collects the data will be useful to the reader
  • Lines 49-75 look like the belong in the Methods section (Section 2)
    • Can also shorten these lines if needed
  • Could the authors explain why the strong reflections of the lunar subsurface materials can't be used to build the lunar velocity models? 
    • I assume that the lunar subsurface materials of interest also generate strong reflections. Is that not the case?

Section 2

  • Could the authors explain why diffractions have non-planar local slopes?
  • Typo in line122: 'flirters' used instead of 'filters'

Author Response

Thanks for your helpful suggestions, we have  revised the manuscript according to your comments and reposoned to your comments. Please refer to the attachment for details.

Reviewer 2 Report

The manuscript entitled “Velocity analysis using separated diffractions for lunar penetrating radar obtained by Yutu-2 rover” reports an interesting analysis of the lunar penetrating radar aboard Yutu-2 rover. The authors clearly describe the method adopted in this study, providing a validation test based on synthetic data. The results are exhaustively presented and are of great interest to the science community. In the attached file, I reported my main comments regarding paragraphs that should be expanded or modified to improve the clarity of the manuscript. I recommend publication of the manuscript after consideration be given to the comments reported in the attached file.

Author Response

(The authors gave the same response as above.)
